# Overlapping Neural Responses to Reflectional Symmetry and Glass Patterns Revealed by an ERP Priming Paradigm

John Tyson-Carr *, Giulia Rampone , Elena Karakashevska , Yiovanna Derpsch , Marco Bertamini † and Alexis D. J. Makin

Department of Psychological Sciences, University of Liverpool, Liverpool L69 3BX, UK; giulia.rampone@liverpool.ac.uk (G.R.); e.karakashevska@liverpool.ac.uk (E.K.); y.derpsch@uea.ac.uk (Y.D.); marcob@liverpool.ac.uk or marco.bertamini@unipd.it (M.B.); mbrxgam4@liverpool.ac.uk (A.D.J.M.)
* Correspondence: hljtyson@liverpool.ac.uk
† Current Address: Department of General Psychology, University of Padova, 35131 Padova, Italy.

**Abstract:** The extrastriate visual cortex is activated by visual regularity and generates an ERP known as the sustained posterior negativity (SPN). Spatial filter models offer a biologically plausible account of regularity detection based on the spectral properties of an image. These models are specific to reflection and therefore imply that reflectional symmetry and Glass patterns are coded by different neural populations. We utilised the SPN priming effect to probe representational overlap between reflection and Glass patterns. For each trial, participants were presented with a rapid succession of three patterns. In the Repeated condition, three reflections or three Glass patterns were presented. In the Changing condition, patterns alternated between reflection and Glass patterns. An increase in SPN amplitude (priming) was observed in both the Repeated and Changing conditions. Results indicate a greater representational overlap in the brain between reflection and Glass patterns than predicted by spatial filter models.

**Keywords:** sustained posterior negativity; SPN priming; reflectional symmetry; glass patterns; extrastriate cortex



## 1. Introduction

In reflectional symmetry, elements on one side of an axis are matched on the other side. For the case of a single axis, we can find many examples in nature, for example, in the overall mirror symmetry of the human body. The neural response to reflectional symmetry has been studied extensively. Functional MRI studies consistently find that reflectional symmetry activates a network of extrastriate regions, such as V3a, V4, and the Lateral Occipital Complex (LOC), but not V1 or V2 [1–6]. Moreover, TMS studies have found that the disruption of the right LOC slows the discrimination of reflectional symmetry [7]. Finally, EEG studies have found that the extrastriate symmetry response generates an ERP called the sustained posterior negativity (SPN) [8–12].

Glass patterns are a different kind of visual regularity. They are produced when a random arrangement of dots is superimposed on itself, and then it is slightly offset [13,14]. Psychophysical studies have investigated the discrimination of radial, concentric, and translational Glass patterns [15–17]. Like reflectional symmetry, Glass patterns activate V4 and LOC [18,19]. In addition, lesions to V4 impair Glass pattern perception [20]. The steady-state VEP response to reflectional symmetry and Glass patterns is comparable [21,22], and both regularities generate the same SPN [12,23]. SPNs generated by reflection and Glass patterns are shown in Figure 1A.

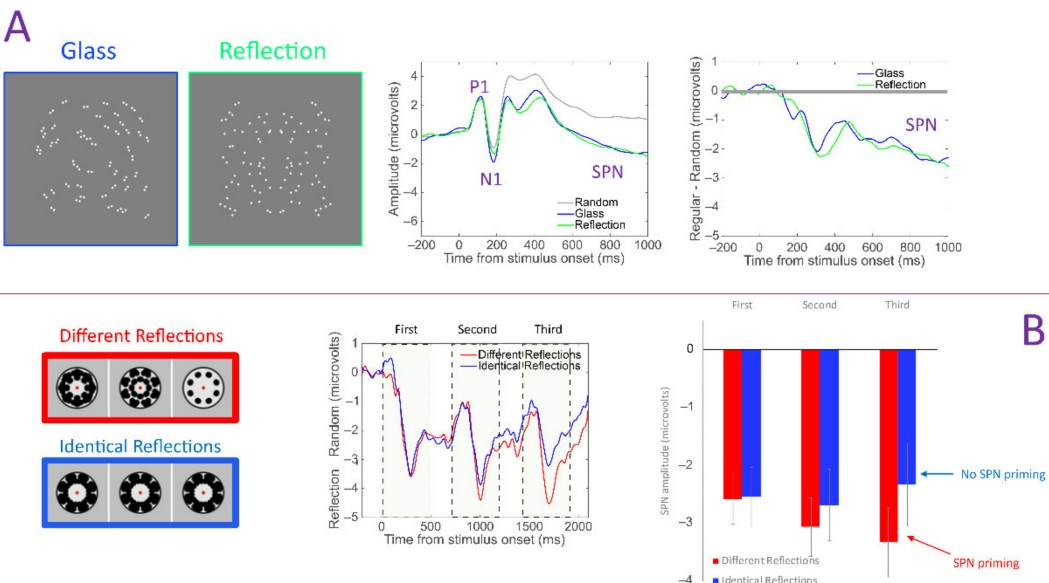

**Figure 1.** ERP response to reflectional symmetry and Glass patterns: (**A**) Results from Experiment 2 of Makin et al. [12]. The left panel shows an example of reflection and Glass patterns. The middle panel shows ERP waves generated by these patterns. The right panel shows the SPN difference wave (regular–random). Reflection and Glass generated very similar SPN waves. (**B**) SPN priming effect in the work by Makin et al. [24]. Repeated presentation of different reflectional symmetries leads to increase in SPN amplitude (SPN priming, red). Repeated presentations of identical exemplars do not lead to increases in SPN amplitude (no SPN priming, blue). The current study examines where the SPN priming effect transfers from reflection to Glass patterns.

In summary, reflection and Glass patterns may be represented in similar perceptual systems in the extrastriate cortex. This is not surprising; both are visual gestalts, which can only be detected by integrating a local structure across the visual field. Furthermore, according to formal models, reflection and Glass patterns are both holographic regularities with near-identical configural goodness [25].

The neural system that underpins the processing of these regularities, which are reflectional or Glass patterns, can be understood as a series of processes operating on increasingly complex information. Low-level vision can be modelled as a retinotopic array of orientation and frequency-tuned filters [26]. Mid-level vision, which mediates global form perception, requires the pooling of signals arising from many low-level filters [17,27–31]. However, Glass patterns and reflectional symmetry have different spectral properties and require different global pooling operations. They also differ in ecological significance. Phenotypic reflectional symmetry guides mate selection in many species [32,33]. Moreover, Glass patterns do not resemble faces or bodies, although they are still visually pleasing [34]. Reflectional symmetry is also a property of many whole objects [35–41], whereas Glass patterns are not related to objecthood, except in the general sense of a non-accidental property of the scene.

Given these similarities and differences between reflections and Glass, we investigated the extent of overlap in the neural responses. Functional overlap and independence can be assessed with repetition paradigms. Repeated presentation usually leads to changes in the neural response [42,43]. If stimulus A and B are coded by independent neural networks, then repetition effects do not transfer from stimulus A to B. Conversely, if stimulus A and B are coded by the same neural networks, the presentation of stimulus A alters the subsequent response to B. Krekelberg et al. [18] used this repetition paradigm to demonstrate the independence of concentric and radial Glass patterns.

Utilising this priming paradigm, SPN priming effects have been reported. Makin et al. [44] recently demonstrated that the repeated presentation of reflectional symmetry leads to an

increase in SPN amplitude. They termed this effect SPN priming. An interesting finding is that SPN priming happens for the repeated presentation of different reflection exemplars, but not for identical exemplars (Figure 1B). Moreover, SPN priming did not transfer across retinal locations or unpredictable changes in axis orientation. However, SPN priming did transfer from a vertical to a horizontal reflection as well as between reflectional and rotational symmetry. In another study, we found that SPN priming transferred between black and white exemplars [24]. SPN priming is consistent with behavioural results, where the repeated presentation of different exemplars can enhance symmetry detection [45].

In the current experiment, we present trials with three regularities in sequence (Figure 2). As in the work conducted by Makin, Piovesan, Tyson-Carr, Rampone, Derpsch, and Bertamini [24], participants completed an oddball detection task, where they noted infrequent trials where the second pattern was blank (Figure 3). We confidently predicted that we would see SPN priming on repeated regularity sequences (Ref > Ref > Ref or Glass > Glass > Glass). However, we tentatively predicted that SPN priming would not survive changing regularity (Ref > Glass > Ref or Glass > Ref > Glass), given the putative differences between these regularities. The study and predictions were pre-registered (https://aspredicted.org/4u583.pdf; accessed on 13 November 2019).

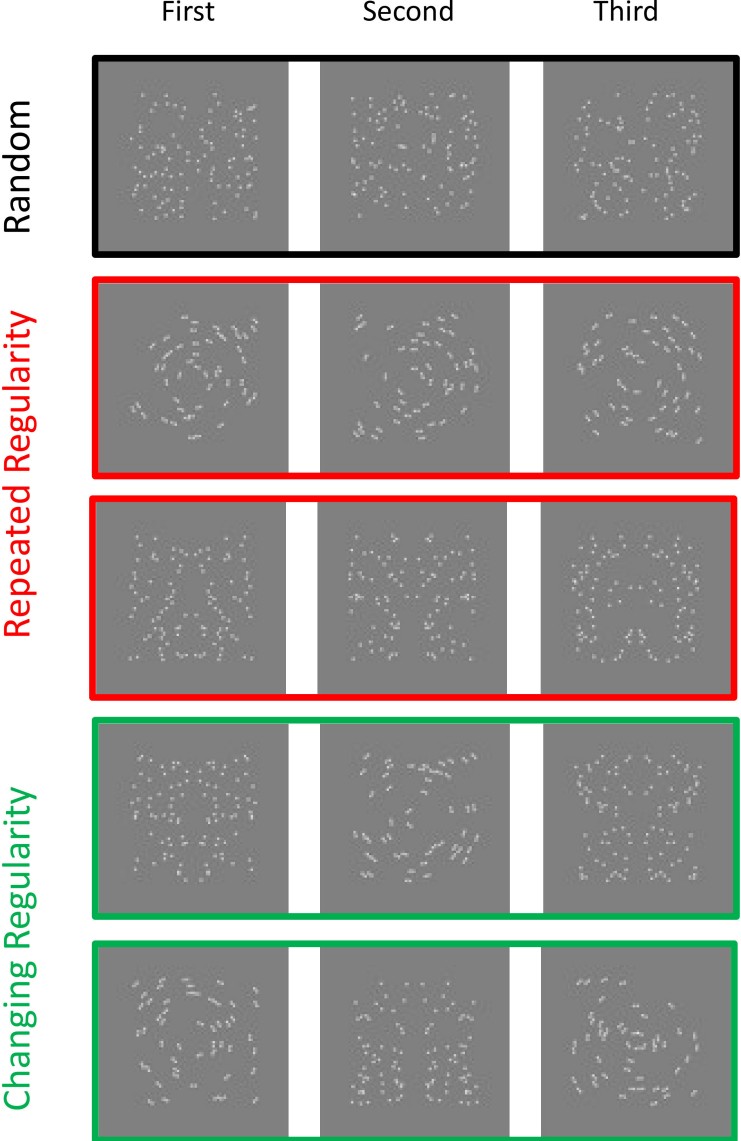

**Figure 2.** Example triplet sequences from the random, repeated regularity and the changing regularity conditions.

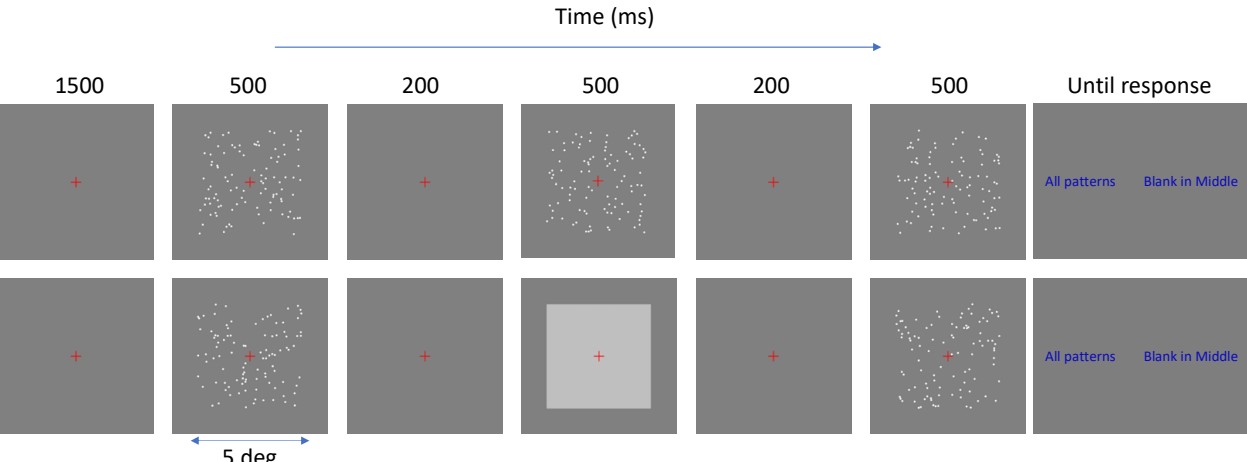

**Figure 3.** Trial structure. In 480 trials (89%), participants were presented with a sequence of three dot patterns (500 ms) separated by 200 ms gaps (top row). In just 60 trials (11%), there was a grey square in the second sequence position (bottom row). The participant's task was to classify trials as 'All patterns' or 'Blank in the middle'.

## 2. Materials and Methods

Twenty-four participants were involved (their ages were 18–24, 12 were male, and 2 were left-handed). Participants had normal or corrected to normal vision. The study had the local ethics committee's approval and was conducted in accordance with the 2008 Declaration of Helsinki. Informed consent was obtained for each participant prior to completion of the study.

Participants were positioned 57 cm from a 29 × 51 cm LCD monitor with a 60 Hz refresh rate. A chin rest was used for gaze stabilisation. EEG data were recorded continuously at 512 Hz from 64 scalp electrodes (BioSemi Active-2 system, Amsterdam, The Netherlands). Horizontal and vertical EOG external channels were used to monitor excessive blinking and eye movements. The experiment was programmed in Python using open-source PsychoPy libraries [46]. The experiments, stimuli, and raw pre-processed EEG data are available at the Open Science Framework. This is project 21 in our complete Liverpool SPN catalogue, an online repository of all SPN research from the University of Liverpool (https://osf.io/2sncj/; accessed on 7 October 2020).

The dot pattern stimuli were the same as those used in the work by Makin et al. [12], where these reflection and Glass patterns were found to produce near-identical SPNs (Figure 1A). The patterns were approximately 5 degrees wide. Each comprised 100 dots. Temporal parameters were the same as that which was used in the work by Makin et al. [44]. In each trial, we presented three dot patterns. Each pattern was presented for 500 ms. There was a 200 ms gap between each pattern (Figure 3).

In 50% of the trials, all three patterns were random. In 25% of the trials, the triplet sequences involved the same regularity type (Ref > Ref > Ref, or Glass > Glass > Glass). In 25% of the trials, the triplet sequences involved changing regularity types (Ref > Glass > Ref or Glass > Ref > Glass). There were 480 trials in total, with 240 random sequences and 60 repeats of each of the regularity sequences. There were an additional 60 trials, which had a blank grey square in the middle (Random Blank Random). The participant's task was to judge whether all three presentations in the trial were patterns (all patterns) or whether there was a blank square in the middle (blank in middle). The blank-in-the-middle oddball trials were not included in any ERP analysis. The task was trivial, so participants gave the correct response in >95% of trials. Incorrect response trials were included in ERP analysis.

The experiment was divided into 30 blocks of 18 trials. Each block had two Random > Oddball > Random trials and an equal distribution of all conditions in the experiment. The trials within a block were presented in a novel, randomised order for each participant. The stimuli were created offline and were saved as PNG files. Every participant saw

the same set of images without repetition. The stimulus images were shuffled; therefore (for example), a particular Glass pattern would be the first in a Glass > Glass > Glass sequence for one participant and the third in a Glass > Ref > Glass sequence for another. One additional block at the start of the experiment served as practice.

Pre-processing conventions were chosen a priori and were pre-registered on aspredicted.org. EEG data from 64 channels were analysed offline using the EEGLAB 13.3.4b toolbox in Matlab [47]. The data were re-referenced to the scalp average, low-pass filtered at 25 Hz, downsampled to 128 Hz, and segmented into $-0.5$ to $+2.1$ s epochs with a $-200$ ms pre-stimulus baseline. Eye blinks and other large artefacts were removed using Independent Components Analysis (Jung et al., 2000). An average of 9.41/64 components were removed per participant, and 8–9% of trials where the amplitude exceeded $+/-100$ microvolts were excluded (worst participant trial exclusion rate = 36%).

The SPN was defined as the difference between reflection and random waves at an a priori posterior electrode cluster [PO7, O1, O2, and PO8]. Therefore, SPN in the Repeated regularity condition was defined as Repeated regularity–random (averaging over Ref > Ref > Ref and Glass > Glass > Glass sequences). SPN in the Changing regularity condition was defined as Changing regularity–random (averaging over Ref > Glass > Ref and Glass > Ref > Glass sequences). Three windows were chosen for statistical analysis of SPN priming: the first window = 250–600 ms, the second window = 950–1300 ms, and the third window = 1650–2000 ms. These windows were defined a priori based on previous SPN priming research [24,44]. These windows were selected in order to capture the primary time window where the SPN was present following each stimulus presentation. The 250–600 ms interval after stimulus onset represents the maximum of the SPN, and thus, the three time windows represent this interval following each stimulus presentation. SPN was analysed with repeated measures ANOVA. There were 2 within-subjects factors [Sequence position (first, second, third) $\times$ 2 Sequence type (repeated regularity, changing regularity)]. The Greenhouse–Geisser correction factor was applied when the assumption of sphericity was violated (Mauchly's test $p < 0.05$).

## 3. Results

ERPs from the posterior electrode cluster are shown in Figure 4. Figure 4A collapses over the regularity type. Unexpectedly, there was an SPN priming effect in both the Repeated and Changing regularity sequences. This was confirmed by a main effect of Sequence position (F (1.232, 28.340) = 9.028, $p = 0.003$, partial $\eta^2 = 0.282$). There was no main effect of the Sequence type (F < 1) and no Sequence position X Sequence type interaction (F (2,46) = 2.470, $p = 0.096$). All six SPNs in Figure 4A represent a significant response to regularity (amplitude < 0, according to one sample $t$ test, $p < 0.001$, minimum effect size Cohen's d = 0.792).

Figure 4B,C separate the responses to reflection and Glass patterns. These results can be characterised by interactions. However, these are all driven by the fact that the reflection SPN was significantly greater than the Glass SPN (t (23) = 3.774, $p = 0.001$, d = 0.770). This contrasts with the results of Makin et al. (2016) [12], where these patterns generated a similar SPN (Figure 1A).

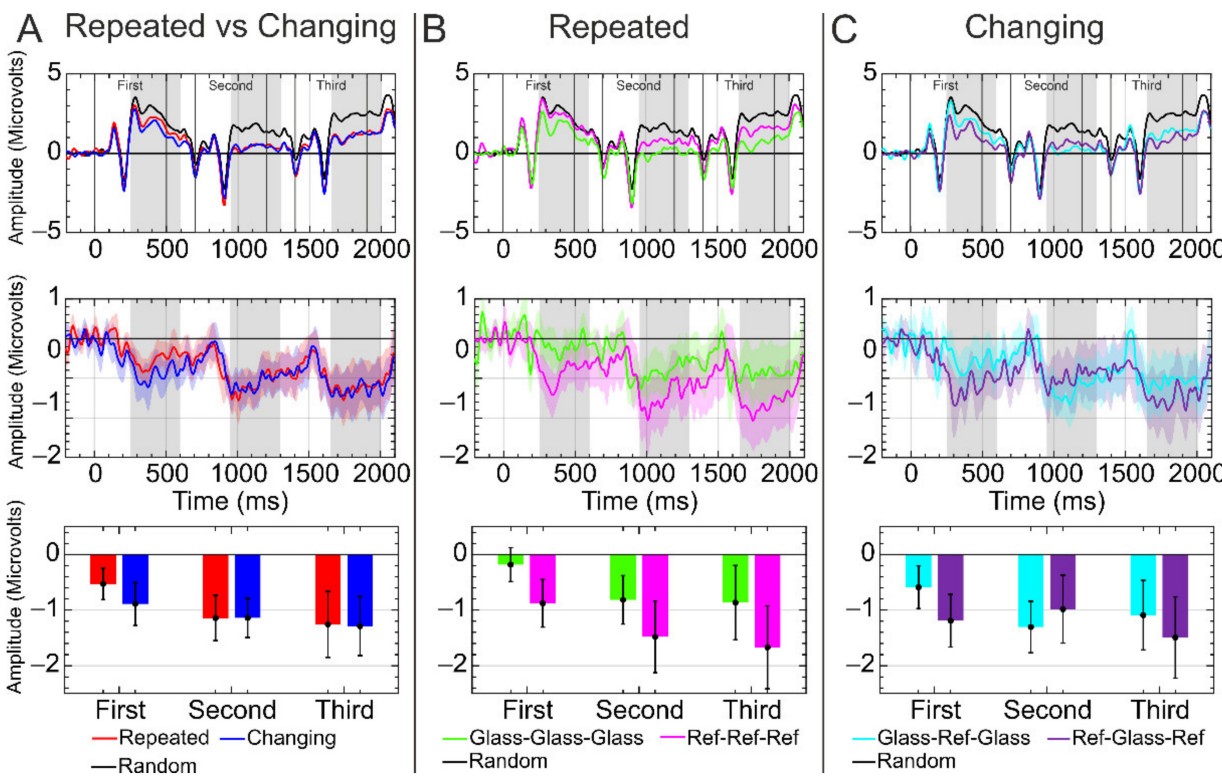

**Figure 4.** ERP results: (**A**) Waves from the Repeated vs. Changing regularities. Top panel shows grand average ERPs from the posterior electrode cluster [PO7, O1, O2, and PO8]. Middle panel shows the SPN as a difference wave. Lower panel shows the average SPN in first, second, and third intervals (250–600, 950–1300, and 1650–2000 ms). Error bars represent +/−95% CI (the fact that no error bar crosses 0 for the Repeated vs. Changing figure indicates a significant SPN in all conditions). (**B**) Waves from the Glass > Glass > Glass (green) and Ref > Ref > Ref (pink) repeated sequences. (**C**) Waves from the Ref > Glass > Ref (cyan) and Glass > Ref > Glass (purple) changing sequences. Note that reflection generates a larger SPN than Glass in (**B**,**C**) (effect is reversed in the middle of the changing sequences). The grey shading in the top and middle panels illustrates the time windows that were analysed. Vertical lines in the top panel refer to the stimulus presentation window.

## 4. Discussion

Contrary to our pre-registered predictions, the SPN priming effect was equivalent in the Repeated and Changing regularity conditions. In other words, the SPN priming effect survives a change from reflection to Glass patterns. This unexpected result suggests that there is overlap between the neural coding of reflection and Glass patterns in the extrastriate cortex. This adds to previous studies which have shown the transfer of SPN priming from black to white patterns [24] and from rotational to reflectional symmetry (Experiment 5 of Makin et al. [24]). Despite the presence of SPN priming effects in previous work, the transfer of SPN priming between reflection and Glass patterns is surprising. Reflectional symmetry plays an important role in mate selection [32] and in facilitating both face perception [48] and figure–ground segmentation [41,49]. Glass patterns are not involved in any of these things. Despite these visual differences, both reflection and Glass patterns seem to be processed by overlapping neural mechanisms.

Filter models have been proposed for reflectional symmetry [30,31,50,51], and these models may represent the processes that neural mechanisms utilise to process reflectional symmetry. One early account is the blob-alignment model [27,52]. Low-pass filtering of any image containing reflectional symmetry results in parallel, mid-point colinear 'blobs' that are all centred on, but perpendicular to, the axis of reflection. Blob alignment can then be used to detect reflection. This is biologically plausible, because early vision can be characterised as an array of orientation and frequency-tuned filters [26]. Related but

different filter models have been suggested for Glass patterns. Wilson et al. [53] proposed a model that pools orientation-selective responses in V1 and that is then sensitive to the quasi-circular nature of concentric Glass patterns (rather than midpoint colinear blobs). This is supported by behavioural and neural evidence [17,19,20,54–56]. Implied motion may also be selectively evoked by Glass patterns [18] but not by reflection. Given the differences between reflection and Glass patterns, we expected no overlap between their neural representations. However, SPN priming suggests that there is, in fact, some overlap.

The liberal transfer of SPN priming has been previously observed. For example, SPN priming was observed between symmetrical patterns with opposite luminance polarities [24] and also between reflection/rotational and vertical/horizontal symmetry when presented in the same retinotopic location [44]. If SPN priming always happened without fail, it could be parsimoniously interpreted as a top-down attentional effect. We know that attention to regularity can enhance SPN amplitude [24], and sequential presentation can simply attract attention to regularity. However, SPN priming is not completely ubiquitous. When three identical reflections are shown, there is no SPN priming (Figure 1B). Likewise, there is no SPN priming when reflectional symmetry changes between unpredictable oblique orientations or between left and right hemifields [44].

It could be argued that the SPN is not an index of active regularity processing, such as the pooling of information across frequency-tuned filters. It is possible that the SPN should not be considered a 'processing component', but an index of stable, finished gestalt representations that have already been processed. Indeed, more sophisticated latency analysis in Kohler et al. [3] suggests that regularity processing begins as early as 75 post-stimulus onset in V3 and V4, much before the SPN. It is possible that different filtering and pooling operations are recruited for reflection and Glass patterns, and these operations can be completed within the first 200 ms, before SPN onset. However, there does seem to be some representational overlap during the SPN interval. It is unclear why these different early processing operations go on to produce overlapping representations at a later stage.

Finally, we note that reflectional symmetry generated a larger SPN than Glass patterns. This indicates that there are some perceptual differences, even though priming transferred. Previous work has reported a similar SPN between reflectional symmetry and Glass patterns [12], and this is not surprising given that these patterns have near-identical perceptual goodness as defined by the holographic weight of evidence model [25]. However, in contrast to previous work, where participants actively discriminated regularity [12], the current study instead utilises an oddball discrimination task. It is possible that reflection SPN is less task-sensitive than the Glass SPN. Perhaps the response to Glass patterns is selectively attenuated when participants are not attending to regularity. Glass patterns do not have a special ecological significance, so the response may be less automatic and robust to task manipulations.

## 5. Conclusions

Similar to previous work [44], the current paper demonstrates a similar SPN priming effect across reflectional symmetry patterns. In contrast to the expected results, SPN priming also transferred between reflection and Glass patterns. The transfer of SPN priming between reflection and Glass patterns describes a set of visual areas in the extrastriate cortex that specialise in extracting regularities from visual input, rather than tuning in to specific types of regularity.

**Author Contributions:** Conceptualisation, M.B. and A.D.J.M.; Formal analysis, J.T.-C. and A.D.J.M.; Funding acquisition, A.D.J.M.; Investigation, E.K., Y.D. and A.D.J.M.; Methodology, M.B. and A.D.J.M.; Software, G.R.; Writing—original draft preparation, J.T.-C. and A.D.J.M.; Writing—review and editing, J.T.-C., G.R., E.K., Y.D., M.B. and A.D.J.M. All authors have read and agreed to the published version of the manuscript.

**Funding:** This research was funded by the Economic and Social Research Council, grant number ES/S014691/1. The APC was funded by the University of Liverpool.

**Institutional Review Board Statement:** The study was conducted in accordance with the Declaration of Helsinki, and approved by the Ethics Committee of University of Liverpool (protocol code 4086 approved on 4 September 2018).

**Informed Consent Statement:** Informed consent was obtained from all subjects involved in the study.

**Data Availability Statement:** All data and code are available at https://osf.io/2sncj/ (accessed on 7 October 2020).

**Conflicts of Interest:** The authors declare no conflict of interest.

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
