# Peer review of "Overlapping Neural Responses to Reflectional Symmetry and Glass Patterns Revealed by an ERP Priming Paradigm"

_symmetry, doi:10.3390/sym14071329_

Round 1

Reviewer 1 Report

This is pre-registered study, therefore I do not comment on the methods (recoring, data processing, data presentation). However, the authors have to deal with a theoretical issue. I'm not sure that the late negativity is a 'processing component'. This is because a huge literature shows that high level of visual processing is much faster (e.g. facial features like emotion, gender, age is processed within 200 ms, non-primed semantic categories are identified within 200 ms, etc. I'd be curious about CRT results  for symmetry/Gless pattern/reandom patterns discrimination. .My guess is that it is faster than the range of the negativity. Therefore the authors have to discuss the possibility that this component is a signature of post-recpetual effect of gestalt-like, salient stimuli.

Reviewer 2 Report

Overlapping neural responses to reflectional symmetry and Glass patterns revealed by an ERP priming paradigm.

Building on previous literature that investigated how visual regularity is processed in the visual cortex, the authors asked whether two types of regularity, namely reflectional symmetry and Glass patterns, are processed by independent neural networks.

Although sharing similarities, reflectional symmetry and Glass patterns are different enough to hypothesise they might be processed by separate neural networks in the extrastriate visual cortex. The authors used the Sustained Posterior Negativity (SPN) component, an ERP known to be generated by visual regularity as priming to determine the extent to which neural networks that respond to those two types of visual regularity differ. In contradiction to their predictions, the authors found equivalent SPN priming effects for both types of visual regularities, despite changes from one type of regularity to the other, suggesting a greater overlap between networks processing them than expected.

This is an interesting result that brings novel questions related to the processing and perception of regularity patterns and calls for further experiments, e.g. to disentangle the potential role of attention, as briefly highlighted by the authors.

Comments:

First, I would like to share my appreciation for the pre-registration initiative and the lab’s SPN catalogue on OSF.

As a general comment, I feel that both the introduction and discussion sections lack some fluidity in their transitions. The paper would benefit from better articulating the link between the different paragraphs of those sections.

Material and Methods:

- The details regarding the 60 additional trials don’t seem to match between the legend of Figure 3 and the text (line 125) regarding whether the participants saw a grey or a white square.

- line 152: “Three windows were chosen for statistical analysis of SPN priming. […] These windows were defined a priori based on previous SPN priming research [24,44]”

Could the authors indicate what motivated the use of those 3 windows in the first place? I couldn’t find anything also in the given references.

Results:

- Figure 4: should be improved. A bigger font size is needed, especially for the lower panel whose legend is hard to read. A column-wise colour code could be added for simplicity.

Regarding the top-left graph, it is not clear what the horizontal lines represent as they are also not mentioned in the legend. I assume they are the stimulus presentation time windows but find them a bit confusing as they only appear on one graph. A light-colour marking might also be a better alternative.

For the top and/or middle (whatever works best in terms of visibility) pannels, it would be better practice to visualise the 95% confidence intervals, to have some idea of the variability across participants.

Line 170: “the reflection SPN was significantly greater than the Glass SPN”: could the authors add something in the discussion about why they think that is the case? Even if there is no significant effect of priming transfer between types of regularities, the fact that there is a weaker SPN for Glass patterns could indicate some processing differences.

- Line 188: There is a typo, 'shown' should be 'showed'.

Round 2

Reviewer 1 Report

The authors revised the ms. properly. I have no further comments.